# Trends in Hospital Admissions Due to Neoplasms in England and Wales between 1999 and 2019: An Ecological Study

**DOI:** 10.3390/ijerph19138054

**Published:** 2022-06-30

**Authors:** Abdallah Y. Naser, Hassan Alwafi, Sara Ibrahim Hemmo, Hamzeh Mohammad Alrawashdeh, Jaber S. Alqahtani, Saeed M. Alghamdi, Moaath K. Mustafa Ali

**Affiliations:** 1Department of Applied Pharmaceutical Sciences and Clinical Pharmacy, Faculty of Pharmacy, Isra University, Amman 11622, Jordan; ac1012@iu.edu.jo; 2Faculty of Medicine, Umm Alqura University, Mecca 21514, Saudi Arabia; hhwafi@uqu.edu.sa; 3Department of Ophthalmology, Sharif Eye Centers, Irbid 11511, Jordan; dr_hmsr@yahoo.com; 4Department of Respiratory Care, Prince Sultan Military College of Health Sciences, Dammam 34313, Saudi Arabia; alqahtani-jaber@hotmail.com; 5National Heart and Lung Institute, Imperial College London, London SW7 2BX, UK; s.alghamdi18@imperial.ac.uk; 6Greenebaum Comprehensive Cancer Center, University of Maryland, Baltimore, MD 21201, USA; moaath_mustafa@yahoo.com

**Keywords:** England, hospitalisation, neoplasm, United Kingdom, Wales

## Abstract

Objectives: This study aimed to investigate the trends in neoplasm-related hospital admissions (NRHA) in England and Wales between 1999 and 2019. Methods: This is an ecological study using publicly available data taken from the two main medical databases in England and Wales; the Hospital Episode Statistics database in England and the Patient Episode Database in Wales. Hospital admissions data were collected for the period between April 1999 and March 2019. Results: A total of 35,704,781 NRHA were reported during the study period. Females contributed to 50.8% of NRHA. The NRHA rate among males increased by 50.0% [from 26.62 (95% CI 26.55–26.68) in 1999 to 39.93 (95% CI 39.86–40.00) in 2019 per 1000 persons, trend test, *p* < 0.001]. The NRHA rate among females increased by 44.1% [from 27.25 (95% CI 27.18–27.31) in 1999 to 39.25 (95% CI 39.18–39.32) in 2019 per 1000 persons, trend test, *p* < 0.001]. Overall, the rate of NRHA rose by 46.2% [from 26.93 (95% CI 26.89–26.98) in 1999 to 39.39 (95% CI 39.34–39.44) in 2019 per 1000 persons, trend test, *p* < 0.001]. Conclusion: Hospital admission rates due to neoplasms increased between 1999 and 2019. Our study demonstrates a variation in NRHA influenced by age and gender. Further observational studies are needed to identify other factors associated with increased hospital admissions among patients with different types of neoplasms.

## 1. Introduction

Cancer is one of the leading causes of death in the UK and worldwide [1]. Patients diagnosed with cancer often experience challenging treatment courses and suffer from frequent hospitalisations. The treatment and diagnosis of cancer is a significant public health issue, and hospitalisation due to cancer imposes a significant strain on the economy and health authorities [2]. The treatment and diagnosis of cancer is a complicated process for patients and physicians. Patients diagnosed with cancer commonly struggle to understand their disease, the associated symptoms, and the management plan. One of the significant obstacles cancer patients face is the need for elective and emergent hospitalisation. The factors associated with repeated hospitalisations are mainly attributed to comorbidities, late diagnosis, or treatment failure [3]. Moreover, the physical and psychological symptom burden for patients with cancer predicts health care utilisation [4]. Patients diagnosed with cancer and who have low health literacy are at an increased risk of hospitalisation [5].

Nowadays, patients diagnosed with cancer can be treated with different modalities, including chemotherapy, surgery, radiation therapy, targeted therapy, immune therapy, or different combinations of these. These treatments are associated with many side effects and commonly render patients immunocompromised. Because of the complexity of treatment schedules and associated toxicity, patients with cancer face many challenges in taking care of themselves. This is particularly important in patients who have comorbidities [6].

The UK continues to nurture cancer awareness and screening programs. These initiatives have increased the number of patients diagnosed at the early stages and improved survival outcomes [7]. Despite this, there are many late-diagnosed cancer cases. Moreover, many patients are diagnosed with cancer during hospitalisation. Admission rates are important measures for healthcare monitoring and assessing interventions’ outcomes [8]. Admission rates are mainly influenced by the number of new cases of diseases, which is an essential measure of healthcare quality [8]. There is a paucity of studies exploring recent trends in cancer admissions, and most previous studies have focused on specific populations or particular diseases [9,10,11]. In England, Wales, and Northern Ireland, Ostermann et al. reported that the mortality and length of stay have decreased in patients with haematologic and solid cancers who had unplanned ICU admissions between 1997 and 2013. The study identified age, severity of illness, metastasis, and previous admission history as independent risk factors for increased hospital mortality [12]. Therefore, in our study, we aimed to investigate the trends in neoplasm-related hospital admissions (NRHA) in England and Wales between 1999 and 2019. Unlike prior studies, this analysis gives a comprehensive estimate of the trends in admissions related to all types of cancer, with no restrictions on the type of cancer being investigated. This will provide decision makers with a clear picture of the current situation regarding cancer and its hospitalisation trends, allowing them to better plan for its management.

## 2. Methods

### 2.1. Data Sources and Study Population

We conducted a descriptive temporal ecological study using publicly available data taken from the two main medical databases in England and Wales; the Hospital Episode Statistics (HES) database in England and the Patient Episode Database in Wales (PEDW). The HES database provides detailed information on hospital admissions associated with a wide range of health conditions in England, and the PEDW provides similar information related to Wales residents [13,14,15,16,17,18,19]. The HES database’s patient admitted care data includes all NHS-paid admissions to private or nonprofit hospitals. The NHS is expected to fund 98–99% of hospital activity in England [20,21].

Hospital admissions data were collected for the period between April 1999 and March 2019. The HES and PEDW databases contain hospital admissions data for all types of neoplasms for patients in all age categories, including below 15 years, 15–59 years, 60–74 years, and 75 years and above. We identified NRHA using the tenth version of the International Statistical Classification of Diseases (ICD) system. All diagnostic codes for neoplasms (C00–D48) were used to identify all hospital admissions (our health outcome of interest) related to various types of neoplasms in England and Wales. The HES and PEDW databases record all hospital admissions, outpatient visits, and accident and emergency attendances performed at all National Health Service (NHS) trusts and any independent sector funded by NHS trusts. Data for hospital admissions in England and Wales are available from the years 1999/2000 onwards. Available data include patient demographics, clinical diagnoses, procedures, and duration of stay. HES and PEDW data are checked regularly to ensure their validity and accuracy. Admission is defined as an overnight stay at the hospital. The annual admissions rate was estimated by dividing the total admissions rate across the study period by the number of years.

### 2.2. Data Analysis

Annual NRHA rates for the overall population with 95% confidence intervals (CIs) were calculated using the number of hospital admissions related to each type of neoplasm for each age group divided by the mid-year population of the same age group for the same year. This was accomplished by utilising the following confidence interval equation for the population proportion: *p*^ +/− *z** (*p*^(1 − *p*^)/*n*)^0.5^.

To calculate the annual hospital admissions rate for neoplasms, we collected mid-year population data between 1999 and 2019 from the Office for National Statistics. Annual hospital admission rates for males were calculated using the number of hospital admissions related to each type of neoplasm for each age group among males divided by the mid-year population for males of the same age group of the same year. A similar procedure was followed to calculate the annual hospital admission rates for females.

The trend in hospital admissions was assessed using a Poisson regression model with robust variance estimation. A two-sided *p* < 0.05 was considered statistically significant. The chi-squared test was used to assess the difference between the admission rates in 1999 and 2019. All analyses were performed using SPSS version 25 (IBM Corp, Armonk, NY, USA).

## 3. Results

A total of 35,704,781 NRHA were reported in England and Wales during the study period. The NRHA rate among males increased by 50.0% [from 26.62 (95% CI 26.55–26.68) in 1999 to 39.93 (95% CI 39.86–40.00) per 1000 persons, trend test, *p* < 0.001]. Females contributed to 50.8% of the total number of NRHA, accounting for 18,131,660 hospital admissions with an average of 906,583 per year. The NRHA rate among females increased by 44.1% [from 27.25 (95% CI 27.18–27.31) in 1999 to 39.25 (95% CI 39.18–39.32) in 2019 per 1000 persons, trend test, *p* < 0.001].

The total annual number of NRHA for various types of neoplasms rose from 1,404,381 in 1999 to 2,341,187 in 2019, representing an 46.2% increase in the hospital admissions rate [from 26.93 (95% CI 26.89–26.98) in 1999 to 39.39 (95% CI 39.34–39.44) in 2019 per 1000 persons, trend test, *p* < 0.001].

The most common NRHA were in haematologic and lymphoid malignancies (16.9%), followed by gastrointestinal malignancies (16.2%), followed by benign neoplasms excluding benign neuroendocrine tumours (ICD code: D10–D36) (12.7%) (Table 1).

During the past two decades, the highest increase in the rate of hospital admissions was observed in ICD code C97.x “malignant neoplasms of independent (primary) multiple sites”, followed by melanoma and other malignant neoplasms of the skin, and in ICD code C76–C80 “malignant neoplasms of ill-defined, other secondary and unspecified sites” with 215-fold, 1.33-fold, and 1.04-fold, respectively. However, the hospital admissions rate due to malignant neoplasms of the urinary tract decreased by 8.0% (Figure 1).

Considering age group variations in NRHA, the age group 60–74 years accounted for 38.5% of the total number of NRHA, followed by the 15–59 years age group with 33.7%, and then the age group 75 years and above with 24.8%. The NRHA rate among patients aged below 15 years increased by 12.0% [from 492.28 (95% CI 488.00–496.00) in 1999 to 551.28 (95% CI 547.00–555.00) in 2019 per 1000 persons, trend test, *p* < 0.001]. The NRHA rate among patients aged 15–59 years increased by 16.2% [from 1779.10 (95% CI 1774.00–1784.00) in 1999 to 2067.23 (95% CI 2062.00–2072.00) in 2019 per 1000 persons, trend test, *p* < 0.001]. The NRHA rate among patients aged 60–74 years increased by 41.1% [from 7198.38 (95% CI 7179.00–7217.00) in 1999 to 10,157.37 (95% CI 10,138.00–10,176.00) in 2019 per 1000 persons, trend test, *p* < 0.001]. The NRHA rate among patients aged 75 years and above increased by 63.5% [from 7613.03 (95% CI 7587.00–7639.00) in 1999 to 12,449.34 (95% CI 12,420.00–12,478.00) in 2019 per 1000 persons, trend test, *p* < 0.001] (Figure 2).

### 3.1. Neoplasm-Related Hospital Admission Rates by Sex

The NRHA rates were higher among males compared to females except for malignant neoplasms of the breast; malignant neoplasms of the thyroid and other endocrine glands; malignant neoplasms of ill-defined, other secondary, and unspecified sites (ICD codes: C76–C80); in situ neoplasms (ICD codes: D00–D09); and benign neoplasms excluding benign neuroendocrine tumours, which were more common in females (Figure 3a–c).

### 3.2. Neoplasm-Related Hospital Admission Rates by Age Group

The NRHA rates were higher in the older age groups. This includes melanoma and other malignant neoplasms of the skin; malignant neoplasms of digestive organs; malignant neoplasms of mesothelial and soft tissue; malignant neoplasms of respiratory and intrathoracic organs; malignant neoplasms of male genital organs; malignant neoplasms of the urinary tract; malignant neoplasms of ill-defined, other secondary, and unspecified sites; malignant neoplasms of lymphoid, haematopoietic and related tissue; malignant neoplasms of independent (primary) multiple sites; in situ neoplasms; and neoplasms of uncertain behaviour, polycythaemia vera, and myelodysplastic syndromes. On the other hand, malignant neoplasms of bone and articular cartilage were more common in the younger population (below 15 years). Furthermore, hospital admissions for malignant neoplasms of the eye, brain, and other parts of the central nervous system, as well as malignant neoplasms of the thyroid and other endocrine glands, were higher in patients under the age of 15 (Figure 4a–c).

## 4. Discussion

Our study found a significant increase in the rate of NRHA in England and Wales; there was a 46.2% increase in the hospital admissions rate between 1999 and 2019, with an average increase of 2.3% per year. The most common NRHA were malignant neoplasms of lymphoid, haematopoietic, and related tissue, followed by malignant neoplasms of digestive organs and benign neoplasms, excluding benign neuroendocrine tumours. When NRHA were stratified by gender and age, we noticed significant variations, as seen in other studies [22].

Many factors likely explain the increase in NRHA; cancer incidence has increased in the UK by 5% over the last decade and by 12% since the early 1990s [23]. Cancer incidence in the UK is ranked higher than in 90% of the world [23]. Moreover, the incidence is projected to increase in the UK by 2% by 2035 [23]. The age-standardised cancer incidence rate for women in England grew by 4.8% between 2006 and 2016, rising from 516.2 per 100,000 females in 2006 to 541.1 in 2016. Men’s cancer incidence decreased by 1.1% from 671.0 per 100,000 men in 2006 to 663.4 in 2016 [23]. The increase in cancer incidences among women will likely contribute to an increase in NRHA. Simultaneously, we noticed that, despite a decline in cancer incidences among men, the rate of hospitalisations increased. This could be explained by the fact that cancer patients have more advanced cancer cases upon diagnosis, resulting in increased hospitalisations or better cancer treatments, resulting in a higher likelihood of recurrence (and so follow-up treatment). Furthermore, the increase in NHRA in the UK can also be explained by the increase in older age groups. As shown in our study, patients 60 years old and older accounted for almost two-thirds of NRHA and experienced the fastest increase in NRHA. Life expectancy in the UK has increased over the past few decades [24]. Currently, there are approximately 12 million people aged 65 and older. This “ageing” population has a higher incidence of cancer and hence associated complications related to its treatment. Moreover, the elderly tend to have other comorbidities, which increases their risk of hospitalisation and readmission [25]. Comorbidities in patients with cancer predict an increased risk of hospitalisation [26,27]. Manzano et al. showed that patients with gastrointestinal malignancy who had a Charlson Comorbidity Index of more than two had a 52% higher chance of unplanned hospitalisation than patients with a comorbidity index of zero [22]. Nowadays, the UK health sector is experiencing an emerging challenge in the care of cancer patients. The proportion of cancer patients treated for cancer within 62 days has dropped from 87% in 2010 down to an alarming 14% in 2019 in the NHS in England [28,29]. The increase in waiting time reflects the high burden of cancer care on the health sector, likely increasing NRHA. Moreover, the increase in waiting time for cancer care might change the care from elective to emergent, especially in sick patients.

In the last two decades, revolutions in cancer screening, diagnosis, and treatment have improved outcomes for patients with cancer [30]. The addition of targeted therapy and immune therapy to the anti-neoplastic arsenal has improved patients’ survival rates yet introduced a multitude of new side effects and complications, which frequently lead to hospitalisation. The progressive increase in cancer survivors has expanded the pool of patients who remain at risk for readmission and long-term complications related to treatments. The overall survival of patients with cancer has doubled in the UK over the last 40 years [31].

Our results showed that hospital admissions due to haematologic/ lymphoid and digestive neoplasms accounted for most of the NRHA in the study period. Haematologic and lymphoid malignancies are associated with a higher risk of hospitalisation because these diseases are commonly diagnosed and treated in hospitals. The treatment of acute leukaemia, aggressive lymphoma, and associated stem cell transplantation occurs mainly in the hospital setting. Moreover, patients with haematologic malignancies are at high risk of hospitalisation and of in-hospital death [32]. Because the treatments administered for haematologic malignancies tend to be more myelosuppressive than the treatments for solid malignancies, patients with haematologic malignancies are at higher risk for bleeding and infectious complications that frequently result in hospitalisation. Gastrointestinal cancer accounts for the second most common cause of NRHA. This might be because of the increased need for in-hospital treatments such as complex surgeries and treatment-related complications. In the UK, it has been reported that colorectal cancer is among the most common causes of cancer-related admissions [33]. In an audit monitoring the diagnosis and management of bowel cancer in England, it was reported that one in four patients with colorectal cancer was admitted to the hospital [34]. Colorectal cancer is frequently diagnosed at advanced stages when patients are prone to complications [34]. Additionally, the gastrointestinal malignancy burden is the highest in Europe and North America than in other parts of the world [35]. The high burden can be explained, partly, by the increased frequency of hospitalisation seen in patients with gastrointestinal cancers. Different types of screening tests could have influenced the early detection of colorectal cancer among patients. These include colonoscopy, computed tomography (CT or CAT) colonography, sigmoidoscopy, faecal occult blood test (FOBT) and faecal immunochemical test (FIT), double-contrast barium enema (DCBE), and stool DNA tests [36]. Different groups have made different colorectal cancer screening recommendations. The American Society of Clinical Oncology (ASCO) has produced colorectal cancer screening guidelines to help people with an average risk avoid cancer [37]. Both men and women with an average risk of colorectal cancer should begin one of these testing schedules at the age of 50. People with an average risk have never been diagnosed with colorectal cancer and do not have a family history of the disease, inflammatory bowel disease, or an inherited syndrome such as Lynch syndrome. The U.S. Preventive Services Task Force (USPSTF) recommends that people aged 45 to 75 should have regular screenings [37].

The largest increase in the NHRA rate was seen in ICD-10 code C97.x “malignant neoplasms of independent (primary) multiple sites”. This code is used mainly when two or more independent primary malignant neoplasms are documented, none of which clearly predominates. The data of NRHA related to C97.x starts from 2012, coinciding with the introduction of the ICD-10 fourth edition in the UK [36]. The NRHA rate related to melanoma has increased by 1.33 times. The incidence of melanoma has increased by 135% in the UK since the early 1990s [38]. This change in incidence explains, at least partially, the sudden and significant increase in melanoma-related hospitalisations. Furthermore, the increased survival of melanoma patients increased the pool of patients who may require hospitalisation in the coming years. Nowadays, around 90% of patients with melanoma survive ten years or more in England [38]. In 2004, the UK adopted the reprinted ICD-10 2000 (with updates and corrections), which might explain the sudden increase in ICD-10 C43-44 “melanoma and other malignant neoplasms of skin” [36]. The decrease in admissions related to urinary tract cancers is likely driven by the decreasing incidence of bladder cancer. In the UK, bladder cancer incidence has dropped by 42% since the early 1990s [39]. Smoking is the strongest risk factor for bladder cancer, and the rate of smoking has been declining in the UK over the last four decades [40,41]. It is likely that the incidence of bladder cancer declined because of the remarkable decline in smoking incidence.

Although cancer can develop at any age, the elderly population experiences higher cancer rates and are prone to treatment complications. With ageing, cells accumulate mutations, which lead to a neoplastic transformation [42,43,44]. People aged 75 and over contribute to more than 36.0% of all confirmed cancer cases in the UK [23]. In this study, we found that older patients (60 and above) had a higher rate of cancer admissions compared to younger patients. Older patients are at higher risk of comorbidities and they often suffer from more than one medical condition [45]. Also, they are more prone to polypharmacy-related complications and adverse drug reactions, which may increase their risk of hospital admission [46]. Other factors that could explain the increase in older patients’ admission trends are that older patients commonly live alone and therefore lack assistance [47]. As a result, the elderly population afflicted with cancer is vulnerable and experiences a high risk of hospitalisation [22]. In a Surveillance, Epidemiology and End Results (SEER)-Medicare dataset analysis, O’Neil et al. showed that 92% of the population aged 66 years or older and who received chemotherapy were hospitalised at least once for any reason [48]. These previous observations were consistent with the findings of our study; the rate of NRHA has increased by 40% and 64% in the age groups 60–74 and over 74 years, respectively.

The variation in NRHA between age groups is partly explained by the variation in the median age of cancer diagnosis. Because the rates of most cancers increase with age, the incidence of hospitalisation for NRHA is higher in the older age groups. Cancers with a higher incidence in the younger age group have resulted in a higher hospitalisation rate in this age group. The primary peak of osteosarcoma and Ewing sarcoma occurs in children between 0 and 24 years [49]. This explains the higher incidence of bone and cartilage-related neoplasm admissions in the younger age groups.

Our analysis showed that the increase in NRHA in males was 50% compared to 44% in females. This variation could be explained partly by the higher absolute number of cancer diagnoses in males [23]. The male gender is a risk factor for developing cancer as a previous study reported that males are at a higher risk of developing 32 of 35 tumour sites [50]. In addition, males have a higher risk of developing cancer due to increased exposure to carcinogens including smoking and occupational exposures. Moreover, endogenous biological factors can increase cancer incidence in males [51]. Furthermore, male patients are more likely than females to suffer from cardiovascular disease and have more comorbidities [52]. Male patients also tend to smoke more often and smoking is a significant risk factor for hospital admission. Because breast and thyroid cancer incidences in females are 100 times and 2.5 times higher than in males, this explains the higher incidence of readmission specific to these cancers encountered in females.

In this study, we highlight an important and significant increase in NRHA in England and Wales. In the UK, total healthcare expenditure has increased from 6.9% in 1997 to 10% of the gross domestic product in 2018 [53]. The increase in healthcare expenditure imposes economic burdens on governments worldwide. Public health authorities must investigate factors that have contributed to this increase. Understanding the factors contributing to the increase in NHRA provides opportunities to implement policies and early programs to decrease or reverse this trend. Because cancer incidence is not decreasing in the foreseeable future, programs to improve the population’s general health are needed. Smoking cessation, exercise, and healthy eating habits all reduce cancer-treatment-related complications and mortality in cancer patients [54,55]. Moreover, lifestyle changes improve the health of cancer survivors [56,57].

### Strengths and Limitations

To the best of our knowledge, this is the first study to explore trends in the rates of NRHA in England and Wales without restricting the study to specific inclusion/exclusion criteria. Our study provided detailed hospital admission rates for all types of neoplasm stratified by age and gender, describing the hospitalisation profile for this group of patients for a period of 20 years. We assume that our findings are generalisable and represent the population in England and Wales, as the HES and PEDW databases are considered valuable sources of data to monitor the trends in health-related activities over time at NHS hospitals. At the same time, our study has some limitations. We could not identify other risk factors such as comorbidities and polypharmacy that may impact neoplasms-related admissions. The HES and PEDW databases did not provide data on an individual level but rather on the total absolute numbers of admissions stratified by age and gender on a population level. This rendered our ability to adjust for important confounding variables such as age and gender. Despite the importance of using age-standardised rates to compare NRHA rates across years, the nature of the available data (on the population level, not the patient-individual level) did not allow us to account for this issue. Therefore, our findings should be interpreted carefully. Because this study’s reported hospital admissions data includes emergency, readmission, and elective admissions, findings should be evaluated carefully as this could have resulted in an overestimation of the presented admission rates. Other limitations include a lack of information on gender at the age-group level, rural/urban residence, and ethnicity for NRHA data. In addition, the lack of inclusion and exclusion criteria could be another study limitation. Furthermore, we were unable to control for potential biases such as ecological bias, confusion, selection, and classification bias.

## 5. Conclusions

Between 1999 and 2019, hospital admission rates for several types of neoplasms in England and Wales significantly increased. More research is needed to identify the risk factors that are contributing to the rise in NRHA. The rise in neoplasm-related hospitalisation would necessitate government planning for increased hospital capacity and the implementation of several preventative strategies to reduce the incidence rate of various types of neoplasms that are preventable.

## Figures and Tables

**Figure 1 ijerph-19-08054-f001:**
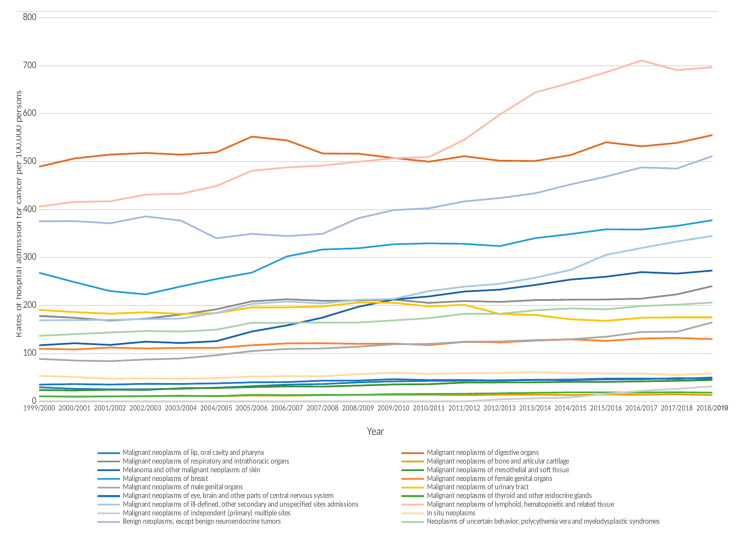
Hospital admission rates due to neoplasms in England and Wales were stratified by type between 1999 and 2019.

**Figure 2 ijerph-19-08054-f002:**
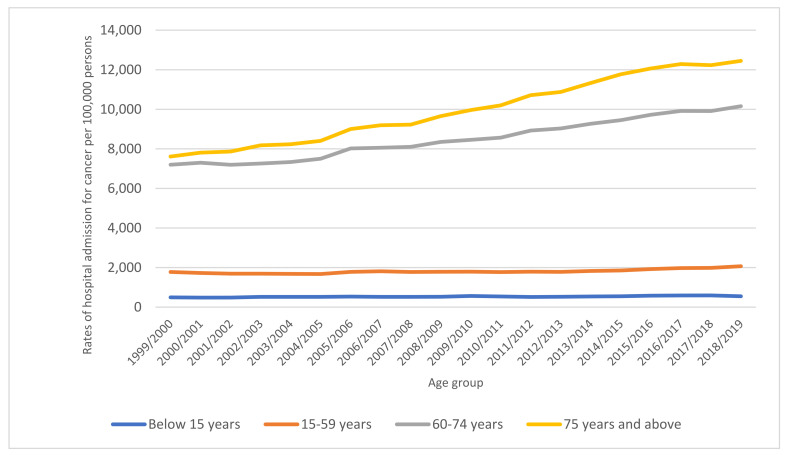
Rates of hospital admission for all neoplasms in England and Wales stratified by age group between 1999 and 2019.

**Figure 3 ijerph-19-08054-f003:**
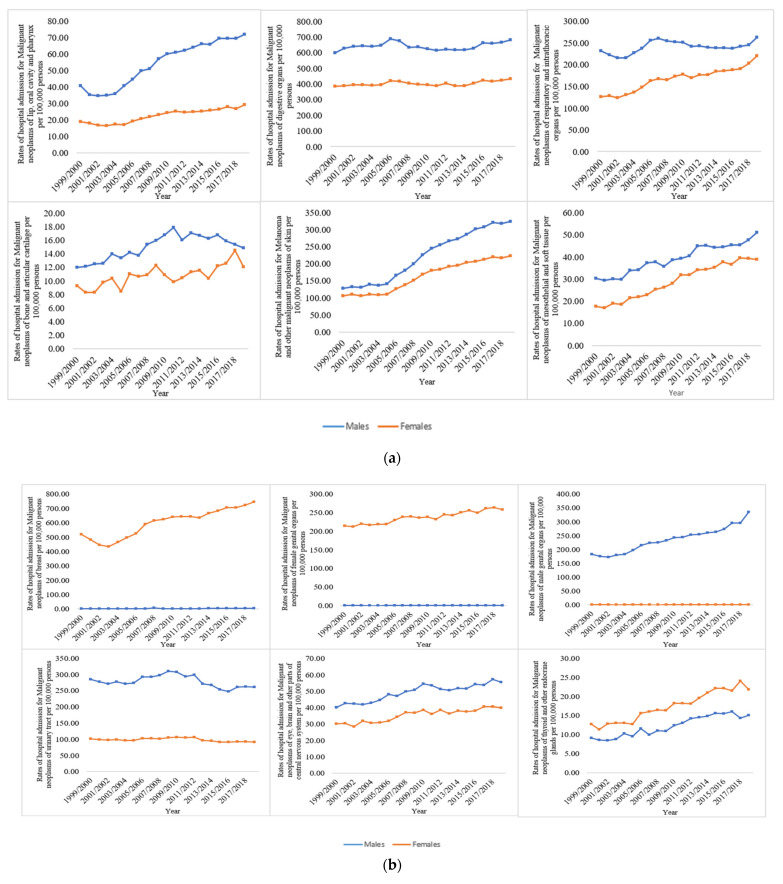
(**a**) Hospital admission rates for neoplasms in England and Wales stratified by gender between 1999 and 2019; (**b**) Hospital admission rates for neoplasms stratified by gender (continued); (**c**) Hospital admission rates for neoplasms stratified by gender (continued).

**Figure 4 ijerph-19-08054-f004:**
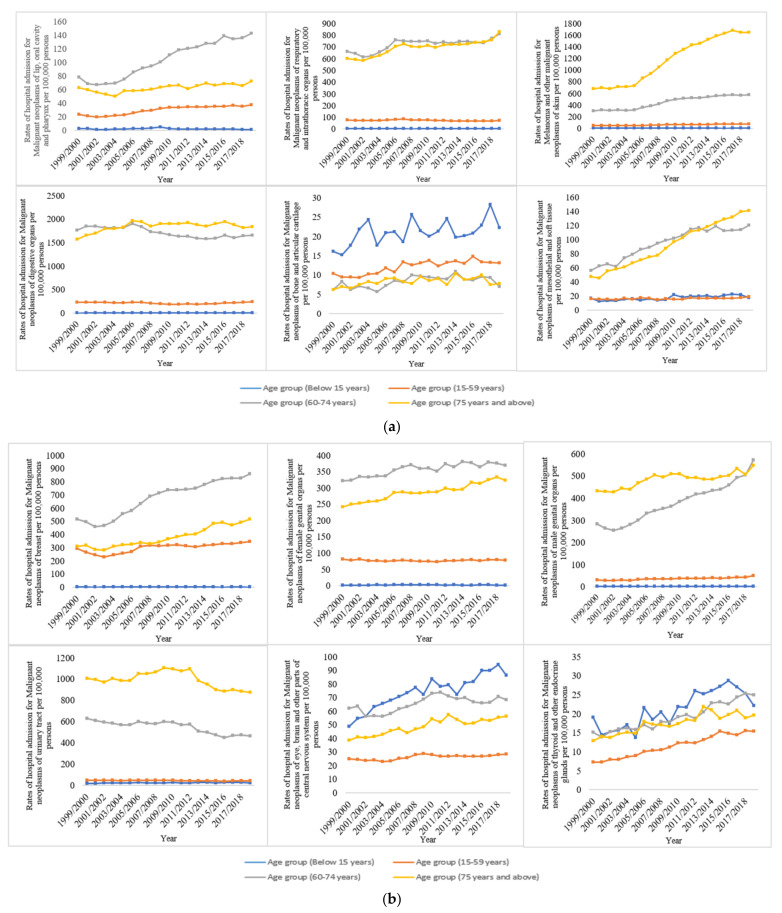
(**a**) Hospital admission rates for neoplasms in England and Wales stratified by age group between 1999 and 2019; (**b**) Hospital admission rates for neoplasms stratified by age group (continued); (**c**) Hospital admission rates for neoplasms stratified by age group (continued).

**Table 1 ijerph-19-08054-t001:** Percentage of neoplasm-related hospital admissions from total number of admissions per ICD code.

ICD Code	Description	Percentage from the Total Number of Admissions
C00–C14	Malignant neoplasms of lip, oral cavity, and pharynx	1.2%
C15–C26	Malignant neoplasms of digestive organs	16.2%
C30–C39	Malignant neoplasms of respiratory and intrathoracic organs	6.3%
C40–C41	Malignant neoplasms of bone and articular cartilage	0.4%
C43–C44	Melanoma and other malignant neoplasms of skin	6.1%
C45–C49	Malignant neoplasms of mesothelial and soft tissue	1.1%
C50–C50	Malignant neoplasms of breast	9.6%
C51–C58	Malignant neoplasms of female genital organs	3.8%
C60–C63	Malignant neoplasms of male genital organs	3.6%
C64–C68	Malignant neoplasms of urinary tract	5.8%
C69–C72	Malignant neoplasms of eye, brain, and other parts of central nervous system	1.3%
C73–C75	Malignant neoplasms of thyroid and other endocrine glands	0.5%
C76–C80	Malignant neoplasms of ill-defined, other secondary, and unspecified sites	7.3%
C81–C96	Malignant neoplasms of lymphoid, hematopoietic, and related tissue	16.9%
C97.X	Malignant neoplasms of independent (primary) multiple sites	0.2%
D00–D09	In situ neoplasms	1.7%
D10–D36	Benign neoplasms, except benign neuroendocrine tumours	12.7%
D37–D48	Neoplasms of uncertain behaviour, polycythaemia vera, and myelodysplastic syndromes	5.3%

ICD International Statistical Classification of Diseases system.

## Data Availability

Publicly available datasets were analyzed in this study. This data can be found here: http://www.infoandstats.wales.nhs.uk/page.cfm?pid=41010&orgid=869 (accessed on 2 February 2022).

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
