# Peer review of "Trends in Hospital Admissions Due to Neoplasms in England and Wales between 1999 and 2019: An Ecological Study"

_ijerph, 2022, doi:10.3390/ijerph19138054_

Round 1
Reviewer 1 Report
With regard to keywords, the main question is whether, according to the authors, they should only concern the spatial scope of the research conducted?
Figures - what area (country) the research concerns (e.g. Fig. 2, 3, 4)?
Author Response
Response to reviewers
Manuscript ID ijerph- 1750014 entitled " Trends of Hospital Admissions due to Neoplasms in England and Wales between 1999 and 2019: An Ecological Study"
Corresponding author: Dr. Abdallah Y Naser
Dear Editor,
Thank you for the opportunity to revise and resubmit our manuscript again based on the reviewers’ comment. Please find below our itemized point-by-point responses to the reviewers’ comment. Answers are written below and edited text has been highlighted (as tracked changes) in the main manuscript.
Reviewer 1:
- With regard to keywords, the main question is whether, according to the authors, they should only concern the spatial scope of the research conducted?
- Thank you for this comment. Actually, we tried to reflect the focus of our research in term of the location and the outcome in our keywords to make it easier for the reader to find out it. However, if the reviewer has any suggestions we are happy to amend them.
- Figures - what area (country) the research concerns (e.g. Fig. 2, 3, 4)?
- Thank you for this comment. All the figures in the manuscript are presenting the findings for England and Wales. We have now highlighted this further in the titles of the figures.
Reviewer 2 Report
Dear authors,
Please, move the sentence in 96-97 lines to “Data Analysis” or integrate it in the paragraph in lines 104-107.
Please, review again the references cite style. Where there are several authors, some are like the first author and then "et al", and others are written differently.
Author Response
Response to reviewers
Manuscript ID ijerph- 1750014 entitled " Trends of Hospital Admissions due to Neoplasms in England and Wales between 1999 and 2019: An Ecological Study"
Corresponding author: Dr. Abdallah Y Naser
Dear Editor,
Thank you for the opportunity to revise and resubmit our manuscript again based on the reviewers’ comment. Please find below our itemized point-by-point responses to the reviewers’ comment. Answers are written below and edited text has been highlighted (as tracked changes) in the main manuscript.
Reviewer 2:
Dear authors,
Please, move the sentence in 96-97 lines to “Data Analysis” or integrate it in the paragraph in lines 104-107.
- Thank you for this comment. We have now addressed the reviewer comment and moved the sentence to the data analysis section.
Please, review again the references cite style. Where there are several authors, some are like the first author and then "et al", and others are written differently.
- Thank you for this comment. We have now addressed the reviewer comment and applying the MDPI referencing style as indicated in the journal’s submission guidelines.
Reviewer 3 Report
This manuscript investigated the trends in neoplasms-related hospital admissions (NRHA) in England and Wales between 1999 and 2019. Several interesting results were concluded. There are several major and minor comments listed below.
1. What trend test was used in the Results? Using Poisson model? What Poisson model was used?
2. In Method, chi-squared test was introduced. Could the authors indicate when chi-squared test was used like they did for the trend test?
3. For the Hospital Episode Statistics database in England and the Patient Episode Database for Wales, did they use same data structure/dictionary? How did the authors combine two database? Is there any missing value? Did the authors conduct any data clean before the analysis.
4. Did the authors explore comparisons between England and Wales?
Minor
1. Page 4, line 134, is 215-fold a typo? There is no C97.x in Figure 1.
2. Some gender-specific cancer rate comparisons may not be necessary presented in Figure 3.
3. Only Figure 2 used rate per 1000 persons. Could the authors change it to per 100,000 persons, which was used for other figures?
Author Response
Response to reviewers
Manuscript ID ijerph- 1750014 entitled " Trends of Hospital Admissions due to Neoplasms in England and Wales between 1999 and 2019: An Ecological Study"
Corresponding author: Dr. Abdallah Y Naser
Dear Editor,
Thank you for the opportunity to revise and resubmit our manuscript again based on the reviewers’ comment. Please find below our itemized point-by-point responses to the reviewers’ comment. Answers are written below and edited text has been highlighted (as tracked changes) in the main manuscript.
Reviewer 3:
This manuscript investigated the trends in neoplasms-related hospital admissions (NRHA) in England and Wales between 1999 and 2019. Several interesting results were concluded. There are several major and minor comments listed below.
- What trend test was used in the Results? Using Poisson model? What Poisson model was used?
- Thank you for this comment. We have now addressed and added the Poisson model used in page 4, lines 112-113.
- In Method, chi-squared test was introduced. Could the authors indicate when chi-squared test was used like they did for the trend test?
- Thank you for this comment. The Chi-square test was used to estimate the difference in the admission rate between the baseline year (1999) and the last year (2019). We have now highlighted this further in the method section, lines 114-115.
- For the Hospital Episode Statistics database in England and the Patient Episode Database for Wales, did they use same data structure/dictionary? How did the authors combine two database? Is there any missing value? Did the authors conduct any data clean before the analysis.
- Thank you for this comment. Yes, the two databases have the same data reporting structure and using the same data dictionary. We combined the data directly by adding the admissions have the same code in England to that in Wales. Then, we estimated the admission rates stratified by age and gender. No missing data and there was no need for data cleaning.
- Did the authors explore comparisons between England and Wales?
- Thank you for this comment. Actually, the aim of this study was not to compare England and Wales but to have an overall estimate about the two countries representing the UK concerning neoplasms hospital admissions.
Minor
- Page 4, line 134, is 215-fold a typo? There is no C97.x in Figure 1.
- Thank you for this comment. The 215-fold increase was not a typo but an actual increase as the rate was extremely low in 1999 and increased markedly in 2019. Concerning the code C97.x in figure 1, we have now uploaded another figure with higher resolution to make it clearer.
- Some gender-specific cancer rate comparisons may not be necessary presented in Figure 3.
- Thank you for this comment. As we found admissions incidents for each type of cancer for both genders we estimated the rate for them and therefore, we preferred to present it graphically as well to make it clear to the reader.
- Only Figure 2 used rate per 1000 persons. Could the authors change it to per 100,000 persons, which was used for other figures?
- Thank you for this comment. We have now addressed the reviewer comment.
This manuscript is a resubmission of an earlier submission. The following is a list of the peer review reports and author responses from that submission.
Round 1
Reviewer 1 Report
The paper is of good quality and the topic is of interest for readers. However, I do have some observations/recommendations:
It is worth considering changing keywords that would not only focus on the spatial area of the research.
The titles of the graphic items (except for Fig. 1) have an incomplete description - sometimes the time range (e.g. Tab. 1) and the spatial range (Fig. 2, 3, 4) are missing.
My dissatisfaction is aroused by the modest conclusions from the paper, which in my opinion should be developed.
Reviewer 2 Report
This manuscript aimed to explore trends of hospital admissions due to neoplasm in England and Wales in the last two decades. There are some concerns as follows:
- The major concern is the aging population in United Kingdom, because the proportion of aging population differs greatly during the last two decades, and thus increased cancer survival, it is necessary to use age-standardized rate to compare NRHA rate between different years.
Maddams, J., Utley, M., & Møller, H. (2012). Projections of cancer prevalence in the United Kingdom, 2010-2040. British journal of cancer, 107(7), 1195–1202.
- The importance of this study is not fully mentioned in the introduction section, please elaborate the difference of current study with previous studies.
- The rates of malignant neoplasm of independent (primary) multiples were quite interesting. Most of them were 0 until 2011/2012, was there any reason behind it?
- Readers may not fully understand the NHS system in UK, such as: Is every hospital covered by NHS trusts? Please give more details about NHS system and its coverage to prove that your data is very completed to reveal the national trends.
- In the Results section, there are some numbers are referred to 1999, some numbers are referred to 2019. However, some numbers are without specific time reference. Please check again with the manuscript.
- Please do the spell check and check typos of the whole manuscript carefully again.
Reviewer 3 Report
My main comment about this paper is that it is entirely descriptive in presenting rates of cancer-related hospitalizations over time. While the discussion section speculates as to what might be underpinning some of the trends, there is little attempt by the authors to investigate these dimensions. Fundamentally, cancer hospitalizations would change for two basic reasons – a higher incidence of cancer cases, and increased incidence of hospitalization among a given set of cancer cases. For the latter, this could occur because of more advanced cancer cases at diagnosis (leading to more hospitalizations) or because of better cancer treatment leading to greater likely recurrence (and so follow-up treatment). I understand that the authors have access only to aggregate data but it would presumably be feasible to compare hospitalization rates to cancer incidence rates in order to give some insight into such factors. Similarly, trends in hospitalization could be contrasted with changes in health behaviors such as smoking rates, with changes in the availability of population cancer screening, and so on, as they might relate to particular cancers. Are any aggregate data available on stage at diagnosis? In short, more could be done to increase the value of this paper’s contribution.
There are clearly changes in classification of cancers that have not been accounted for. The figure in the top right corner of p8 (malignant neoplasms of independent multiples) very clearly illustrates some sort of classification change since actual cancer hospitalizations of a given cancer type would not change like this.
Similarly, though not as starkly, on p11 in the top panel (melanoma) there appears to be a discontinuity in that from 1999-2005 there is little change but from 2006 on there is a substantial steady increase. In the discussion the authors note the ‘sudden significant’ increase in melanoma-related hospitalizations. Why is this? is there a similar marked change in melanoma incidence? Has there been some sort of change in classification?
It is unfortunate that only aggregate data by broad age group are available as age-standardized incidence rates would be the norm for reporting these types of results.
Reviewer 4 Report
Dear authors, thank you for allowing me to review your work.
The manuscript “Trends of Hospital Admissions due to Neoplasms in England and Wales between 1999 and 2019: An Ecological Study” is, under my point of view, a decriptive (exploratory) temporal ecological study.
As I see, your measure is the oficial registered neoplasms anual admisions rate in UK and Gales during the 1999 to 2019 period. Authors compare the anual admissions rate (using mid population) magnitude, as ecological variable, between the temporal units (the differents years during that period), using a national database from the NSH as secondary data source.
The manuscript shows a great effort, but it has some flaws that need to be solved, and several ammendments to be addressed before considering publication.
General considerations
In my opinion it is necessary to concrete some questions about the methodology and to re write the discusión epigraph. In this sense, I see redundant sentences” and unstructured paragraphs in “Discusion”. Please review it. I also suggest to structured “Discusión” with sub headings. Also, in “Discusión” I suggest to avoid the repetition of the same data given in “Results”.
Some questions about methods are,
Which health/s indicator/s do authors have studied in the manuscript?
How do you have quantified the asociation between the ecological variable and the independent variables?
Authors have investigated two time related factors effects, period and age. But, why not date of birth (cohort)?, in case it is an anonymized data base.
You mention Poisson model in line 97. In this sense, how do you have quantified the differences in the ecological variable between the successive temporal units to avoid the random influence?
Did authors make bivariant comparison?
What statistical test have been used to compare the rates?
What do authors think is the manuscript relevance?
Furthermore, about the manuscript writing style, I apologize because I am not native but, I have the perception that English would need to be improved throughout. In my opinion, the manuscript would benefit from a throughout proofread by a native English speaker to improve grammar too.
Specific comments. Those are my specific comments,
Abstract:
Line 15. Please, specify that this is a descriptive (or exploratory) ecological study.
Line 18. Please, consider to add how the trend was assessed at the end of Methods section.
Introduction
Line 53. I suppose that this afirmation comes from the author´s experience. I suggest to support it with some reference.
Line 54. Would you please define “admissión” in UK and Gales? Every hospital admission has to be discharge from a bed hospital? Or is enough to be registered when you come into the hospital, regardless you “sleep” almost a night in hospital. This reflexión is in relation to the posibility of patients who are undiagnosed unless they are studied during the hospital stay (with SCAN for example). Sorry, but in my country is different the concept “admisión” (when you get to a health centre and you are registered), and “hospital discharge” (you need to “sleep” almost one night in a hospital bed.
Line 63. Please add the reference (9) at the end of “hospital mortality” to avoid confusion with the next sentence.
Line 64. “Therefore in this study”… I understand that authors refer to their study.
Methods
Besides the questions asked in “General considerations”.
Line 68. Please, specify the type of ecological study (descriptive or exploratory), and if it is a temporal, spacial or mixt ecological study.
Line 74. About the period when authors collected data, I understand that was once in a time, or year by year? How do you discriminate (and recorded) re-admission of the same neoplasm processes is sucesives years, in order to avoid double, triple… analysis? Or the same patients when they are transferred from another ICU, Hospital, or health centre?
From my point of wiew, the lack of inclusion and exclusion criteria could be a serious study limitation.
Line 92. Please, consider to add the statistical test employed to calculate the confidence intervals. And to compare the independent and dependent variables (in case it is applicable).
Line 99. I suggest mentioning how do you control possible bias (ecological bias, confusion, selection and clasification bias), or espurious association (remember the Granger-Weiner rule). For example, some bias could be due to changes from the neoplasm patology clasification along the time (twenty years) or in the diagnostic criteria. Or because of patients who died before the “mid year” modifying the rate. I also wonder if those clinically undiagnosed neoplasm find out after autopsy are included in the study.
As authors mention Poisson model (line 97), I also suggest to specify how much the Poisson´s coefficient was.
Why not a binomial model to asses the trend in those neoplasic processes which are not “rare”? (like breast cáncer).
Results
For me it is difficult to identify the “highest increase in the rate” (line 117) and “decreased” (Line 121) that you mention in Figure 1, and, in general, to discriminate the different colours. I suggest to try another way to ilustrate it.
Line 128. I suggest to begin in new paragraph the sentence which begins with “The NRHA rate among… in order to avoid confusión between percentages (%) and data per 1000 persons.
Line 139. “Neoplasm hospital admission rate by gender”. I suggest “by sex”, because I consider you did not measured gender. It may be that in Engish is the same.
Line 146. “Neoplasm hospital admission rate by age group”. I suggest to relocate after the paragraph where you begin considering age group variations…(line 126).
Discusión
Line 163. Where does come from the data 2.3% per year? Please, explain in “Methods” how did you calculate it. Also, in this line, “NRHA reasons?” what do you want to mean with “reasons”?
Line 168. “Cancer patients are at higher risk of frequent and unscheduled hospitalisation due to 168 refractory symptoms or acute conditions “. This is the reason because I ask authors previously about how do you discriminate readmissions to assure that you are not counting it twice or more times the same neoplasm process.
Line 180. “Cancer incidence in the UK is ranked higher than 90% of the world [19]”? Please, check this data. How do you explain it, if, authors also metion in line 183 that “Life expectancy in the UK has increased over the last several decades”?
Line 189. “Manzano et al…”. I would recolocate this sentence to line 169 after reference 15.
Line 192. Please, give some reference to evidence the sentence. For example, “Screening for Lung Cancer With Low-Dose Computed Tomography (JAMA. 2021;325(10):971-987. doi:10.1001/jama.2021.0377)”.
Lines 200-201, You have here an example of redundant sentence, because is similar to that in line 163. Another example is in line224 “Older patients are at higher risk of comorbidity and they often suffer from more than one medical condition” where authors express the same idea that in lines 186-187.“Moreover, the elderly tend to have other comorbidities, which increases their risk for hospitalization and readmission”.
Line 205 “Moreover, patients with haematologic malignancies at high risk of hospitalization and for in-hospital death”. Is this afirmation for children too? Please specify what age group patients are you speaking about.
Lines 212 “Colorectal cancer is frequently diagnosed at advanced stages when patients suffer from com213 plications”. I suggest to discuss here if preventive programs (occult blood in feces test screening/ preventive colonoscopy) in UK/Gales, are having some influence, supporting it with some reference.
Line 221. “the decreasing incidence of bladder cáncer”. Could you give some factors to explain that decreasing.
Line 230. Please define “SEER” (Surveillance, Epidemiology and End Results).
Line 238, Why talk about sex differences in cardiovascular diseases if cardiovascular is not the aim of your study? I see, more relevant, for example, the lung cancer evolution in females during the last years, which is approaching to the males rate (figure 3.a), long time after the incorporation of women to the smoking habit due to the lung cancer latency period. This is related to “gender” more than to “sex”.
Lines 257-259 I suggest some reference about to support this sentence. You have this one about eating habits
https://doi.org/10.1186/s12916-022-02256-w
Strengths and Limitations
Line 275-276. “Other limitations include a lack of information on gender at the age-group level, rural/urban residence, and ethnicity for NRHA data”.
Authors mention in line 84 that “Available data include patient demographics…”. In this sense, I encourage you to explote some demographics data.
Conclusion
Line 280-282. “Because cancer survivors are increasing and the population is aging in the UK, effective programs are needed to slow down or reverse the concerning trend in NRHA”. It seems to me an authors opinion more than a conclusion of your study.
References
Please, check and review the references, following an uniform style.
Some references don´t have the date when were accesed from internet.
Response to reviewers
Reviewer 1:
The paper is of good quality and the topic is of interest for readers. However, I do have some observations/recommendations:
It is worth considering changing keywords that would not only focus on the spatial area of the research.
- The keywords are as the following: England; Hospitalisation; Neoplasm; United Kingdom; Wales and covers our research topic, however, if the reviewer have any suggestion we can change them.
The titles of the graphic items (except for Fig. 1) have an incomplete description - sometimes the time range (e.g. Tab. 1) and the spatial range (Fig. 2, 3, 4) are missing.
- We have now addressed this comment.
My dissatisfaction is aroused by the modest conclusions from the paper, which in my opinion should be developed.
- Thank you for this comment, we have now rephrased our conclusion in page 17 based on the reviewer comment.
…………..
Reviewer 2:
This manuscript aimed to explore trends of hospital admissions due to neoplasm in England and Wales in the last two decades. There are some concerns as follows:
- The major concern is the aging population in United Kingdom, because the proportion of aging population differs greatly during the last two decades, and thus increased cancer survival, it is necessary to use age-standardized rate to compare NRHA rate between different years.
Maddams, J., Utley, M., & Møller, H. (2012). Projections of cancer prevalence in the United Kingdom, 2010-2040. British journal of cancer, 107(7), 1195–1202.
- Thank you for this valuable comment. We total agree with the reviewer comment concerning the need for the use of age-standardized rate. However, due to the nature of the publically available data that we have (on the population level) which is not on patient-individual level, we can adjust for the age. However, based on the reviewer comment we have now added this point to the limitation section in page 17.
- The importance of this study is not fully mentioned in the introduction section, please elaborate the difference of current study with previous studies.
- Thank you for this valuable comment. We have highlighted the important of our study in comparison to previous studies in the introduction section in page 2, lines 68-72.
- The rates of malignant neoplasm of independent (primary) multiples were quite interesting. Most of them were 0 until 2011/2012, was there any reason behind it?
- Thank you for this comment. We have now highlighted this point in the discussion as the following in page 15, in the second paragraph “The largest increase in NHRA rate was seen in ICD-10 code C97.x “malignant neoplasms of independent (primary) multiple sites”. This code is used mainly when two or more independent primary malignant neoplasms are documented and none of which clearly predominates. The number of NRHA related to C97.x started in 2012, coinciding with the introduction of ICD-10 fourth edition in the UK”.
- Readers may not fully understand the NHS system in UK, such as: Is every hospital covered by NHS trusts? Please give more details about NHS system and its coverage to prove that your data is very completed to reveal the national trends.
- Thank you for this comment. Hospital Episode Statistics (HES) is a database containing details of all admissions, A and E attendances and outpatient appointments at NHS hospitals in England. HES data covers all NHS Clinical Commissioning Groups (CCGs) in England, including: private patients treated in NHS hospitals, patients resident outside of England, and care delivered by treatment centres (including those in the independent sector) funded by the NHS. Hospital Episode Statistics Admitted Patient Care (HES APC) data are collected on all admissions to National Health Service (NHS) hospitals in England. HES APC also covers admissions to independent sector providers (private or charitable hospitals) paid for by the NHS. It is estimated that 98–99% of hospital activity in England is funded by the NHS. We have now addressed this comment in page 2, lines 80-82.
- In the Results section, there are some numbers are referred to 1999, some numbers are referred to 2019. However, some numbers are without specific time reference. Please check again with the manuscript.
- Thank you for this comment. We have now addressed this comment and added the specific time reference for each number in the results section.
- Please do the spell check and check typos of the whole manuscript carefully again.
- Thank you for this comment. We have now checked the whole document based on the reviewer comment.
Reviewer 3:
My main comment about this paper is that it is entirely descriptive in presenting rates of cancer-related hospitalizations over time. While the discussion section speculates as to what might be underpinning some of the trends, there is little attempt by the authors to investigate these dimensions. Fundamentally, cancer hospitalizations would change for two basic reasons – a higher incidence of cancer cases, and increased incidence of hospitalization among a given set of cancer cases. For the latter, this could occur because of more advanced cancer cases at diagnosis (leading to more hospitalizations) or because of better cancer treatment leading to greater likely recurrence (and so follow-up treatment). I understand that the authors have access only to aggregate data but it would presumably be feasible to compare hospitalization rates to cancer incidence rates in order to give some insight into such factors. Similarly, trends in hospitalization could be contrasted with changes in health behaviors such as smoking rates, with changes in the availability of population cancer screening, and so on, as they might relate to particular cancers. Are any aggregate data available on stage at diagnosis? In short, more could be done to increase the value of this paper’s contribution.
- Thank you for this valuable comment. Unfortunately, we do not have data on stage at diagnosis. All available data was analysed and presented in this study. However, based on the reviewer comment we have now compared hospitalization rates to cancer incidence rates in the discussion section to give some insight into such factor, see page 14, lines 211-223.
There are clearly changes in classification of cancers that have not been accounted for. The figure in the top right corner of p8 (malignant neoplasms of independent multiples) very clearly illustrates some sort of classification change since actual cancer hospitalizations of a given cancer type would not change like this.
- Thank you for this comment. We have now highlighted this point in the discussion as the following in page 15, in the second paragraph “The largest increase in NHRA rate was seen in ICD-10 code C97.x “malignant ne-oplasms of independent (primary) multiple sites”. This code is used mainly when two or more independent primary malignant neoplasms are documented and none of which clearly predominates. The number of NRHA related to C97.x started in 2012, co-inciding with the introduction of ICD-10 fourth edition in the UK”.
Similarly, though not as starkly, on p11 in the top panel (melanoma) there appears to be a discontinuity in that from 1999-2005 there is little change but from 2006 on there is a substantial steady increase. In the discussion the authors note the ‘sudden significant’ increase in melanoma-related hospitalizations. Why is this? is there a similar marked change in melanoma incidence? Has there been some sort of change in classification?
- Thank you for this comment. We have now highlighted this point further in the discussion in page 15, lines 286-288.
It is unfortunate that only aggregate data by broad age group are available as age-standardized incidence rates would be the norm for reporting these types of results.
- We totally agree with the reviewer concerning this point. We have now highlighted this limitation in the discussion section in page 17, lines 373-379.
Reviewer 4:
Dear authors, thank you for allowing me to review your work.
The manuscript “Trends of Hospital Admissions due to Neoplasms in England and Wales between 1999 and 2019: An Ecological Study” is, under my point of view, a decriptive (exploratory) temporal ecological study.
As I see, your measure is the oficial registered neoplasms anual admisions rate in UK and Gales during the 1999 to 2019 period. Authors compare the anual admissions rate (using mid population) magnitude, as ecological variable, between the temporal units (the differents years during that period), using a national database from the NSH as secondary data source.
The manuscript shows a great effort, but it has some flaws that need to be solved, and several ammendments to be addressed before considering publication.
General considerations
In my opinion it is necessary to concrete some questions about the methodology and to re write the discusión epigraph. In this sense, I see redundant sentences” and unstructured paragraphs in “Discusion”. Please review it. I also suggest to structured “Discusión” with sub headings. Also, in “Discusión” I suggest to avoid the repetition of the same data given in “Results”.
- Thank you for this valuable comment. We have now addressed the reviewer comment and rephrased all redundant sentences in the discussion, see pages 14-17.
Some questions about methods are,
Which health/s indicator/s do authors have studied in the manuscript?
- It is highlighted in the method section, lines 88-89, that we identified all hospital admission related to various types of neoplasms in England and Wales. We have now clarified further that this is our main outcome of interest.
How do you have quantified the asociation between the ecological variable and the independent variables?
- We did not quantify any association in our study as we were using publically available data which are on the population level not on patient-individual level. This restricted our ability to find association. We only demonstrated the trend of admissions stratified by type, age, and gender.
Authors have investigated two time related factors effects, period and age. But, why not date of birth (cohort)?, in case it is an anonymized data base.
- We used publically available data which are on the population level not on patient-individual level. These data are on the population level with no information related to date of birth (not a cohort study but ecological study using aggregated data).
Did authors make bivariant comparison?
- No we did not due to the nature of the data available (aggregated data).
What statistical test have been used to compare the rates?
- Chi-square test. We have now highlighted this point in page 3, lines 112-1113.
What do authors think is the manuscript relevance?
- Thank you for this comment. There is a paucity of studies exploring recent trends in cancer admissions, and most of the previous studies focused on specific populations or particular diseases. Our study provides a comprehensive estimation for the trend of admission related to all types of cancer with no restriction on the type of cancer being studied like previous studies. This will give the decision maker a clear imagination for the current situation related to cancer and its hospitalisation pattern, which will enable them to plan for its management more effectively. We have now highlighted this further in page 2, lines 68-72.
Furthermore, about the manuscript writing style, I apologize because I am not native but, I have the perception that English would need to be improved throughout. In my opinion, the manuscript would benefit from a throughout proofread by a native English speaker to improve grammar too.
- We have now addressed the reviewer comment and made a thorough proofreading for it.
Specific comments. Those are my specific comments,
Abstract:
Line 15. Please, specify that this is a descriptive (or exploratory) ecological study.
- We have now addressed the reviewer comment in page 2, line 75.
Line 18. Please, consider to add how the trend was assessed at the end of Methods section.
- This point is highlighted in under that data analysis section in page 3.
Introduction
Line 53. I suppose that this afirmation comes from the author´s experience. I suggest to support it with some reference.
- We have now added relevant references as recommended by the reviewer.
Line 54. Would you please define “admissión” in UK and Gales? Every hospital admission has to be discharge from a bed hospital? Or is enough to be registered when you come into the hospital, regardless you “sleep” almost a night in hospital. This reflexión is in relation to the posibility of patients who are undiagnosed unless they are studied during the hospital stay (with SCAN for example). Sorry, but in my country is different the concept “admisión” (when you get to a health centre and you are registered), and “hospital discharge” (you need to “sleep” almost one night in a hospital bed.
- We have now highlighted the definition of admission in our study, which is overnight stay in the hospital, see page 2, line 97.
Line 63. Please add the reference (9) at the end of “hospital mortality” to avoid confusion with the next sentence.
- We have now addressed the reviewer comment.
Line 64. “Therefore in this study”… I understand that authors refer to their study.
- We have now rephrased our sentence to be “Therefore, in our study….”
Methods
Besides the questions asked in “General considerations”.
Line 68. Please, specify the type of ecological study (descriptive or exploratory), and if it is a temporal, spacial or mixt ecological study.
- We have now addressed the reviewer comment in page 2, line 75.
Line 74. About the period when authors collected data, I understand that was once in a time, or year by year? How do you discriminate (and recorded) re-admission of the same neoplasm processes is sucesives years, in order to avoid double, triple… analysis? Or the same patients when they are transferred from another ICU, Hospital, or health centre?
- Unfortunately, due to the aggregate nature of the data that we used in our study “available online” our estimate might include re-admission data, and we are unable to account for this as we do not have data on the individual patient level. We have now highlighted this point in the limitation section, see page 17, lines 373-379.
From my point of wiew, the lack of inclusion and exclusion criteria could be a serious study limitation.
- As we have mentioned above. We were unable to apply any inclusion/exclusion criteria in our study as the available data are aggregated and on the population level. However, based on the reviewer comment, we have now added this point to the limitations, see page 17, lines 381-384.
Line 92. Please, consider to add the statistical test employed to calculate the confidence intervals. And to compare the independent and dependent variables (in case it is applicable).
- We have now mentioned the procedure used to calculate the confidence intervals in the data analysis section, see page 3, lines 104-105. Concerning the second point, this was not applicable.
Line 99. I suggest mentioning how do you control possible bias (ecological bias, confusion, selection and clasification bias), or espurious association (remember the Granger-Weiner rule). For example, some bias could be due to changes from the neoplasm patology clasification along the time (twenty years) or in the diagnostic criteria. Or because of patients who died before the “mid year” modifying the rate. I also wonder if those clinically undiagnosed neoplasm find out after autopsy are included in the study.
- As we have mentioned above. We were unable to control possible bias in our study as the available data are aggregated and on the population level. However, based on the reviewer comment, we have now added this point to the limitations, see page 17, lines 383-384.
Why not a binomial model to asses the trend in those neoplasic processes which are not “rare”? (like breast cáncer).
- Our study aim was to explore the trend of hospital admissions related to all types of neoplasms. We did not want to change the focus of our study to be directed the comparison of admission rates of specific type of cancer. Besides, the nature of the available data restricted our ability to do further analysis other than the ones presented in our manuscript.
Results
For me it is difficult to identify the “highest increase in the rate” (line 117) and “decreased” (Line 121) that you mention in Figure 1, and, in general, to discriminate the different colours. I suggest to try another way to ilustrate it.
- Thank you for this comment. Any technical issue related to the colour and presentation of the table is usually handled by the production office of the journal.
Line 128. I suggest to begin in new paragraph the sentence which begins with “The NRHA rate among… in order to avoid confusión between percentages (%) and data per 1000 persons.
- We have now rephrased the paragraph and addressed the reviewer comment in page 3.
Line 139. “Neoplasm hospital admission rate by gender”. I suggest “by sex”, because I consider you did not measured gender. It may be that in Engish is the same.
- We have now addressed the reviewer comment in page 5.
Line 146. “Neoplasm hospital admission rate by age group”. I suggest to relocate after the paragraph where you begin considering age group variations…(line 126).
- This paragraph is taking about the comparison between admission rates by age group between different types of neoplasms, while the other paragraph is taking about the difference in the overall rate of admission across different age groups, therefore, we prefer to keep it as it is.
Discusión
Line 163. Where does come from the data 2.3% per year? Please, explain in “Methods” how did you calculate it. Also, in this line, “NRHA reasons?” what do you want to mean with “reasons”?
- This rate was calculated by dividing the overall rate by the number of years to give an average annual estimate (clarified in lines 98-99). Concerning the word “reasons” we have now changed to eliminate any confusion for the reader.
Line 168. “Cancer patients are at higher risk of frequent and unscheduled hospitalisation due to 168 refractory symptoms or acute conditions “. This is the reason because I ask authors previously about how do you discriminate readmissions to assure that you are not counting it twice or more times the same neoplasm process.
- As we mentioned above, due to the aggregate nature of the data that we used in our study “available online” our estimate might include re-admission data, and we are unable to account for this as we do not have data on the individual patient level. We have now highlighted this point in the limitation section, see page 17, lines 376-379.
Line 180. “Cancer incidence in the UK is ranked higher than 90% of the world [19]”? Please, check this data. How do you explain it, if, authors also metion in line 183 that “Life expectancy in the UK has increased over the last several decades”?
- Yes, these data are accurate, please see this reference: Cancer Research UK. Cancer incidence statistics. 2/10/2021]; Available from: https://www.cancerresearchuk.org/health-professional/cancer-statistics/incidence#heading-Zero.
Line 189. “Manzano et al…”. I would recolocate this sentence to line 169 after reference 15.
- We addressed the reviewer comment.
Line 192. Please, give some reference to evidence the sentence. For example, “Screening for Lung Cancer With Low-Dose Computed Tomography (JAMA. 2021;325(10):971-987. doi:10.1001/jama.2021.0377)”.
- We addressed the reviewer comment.
Lines 200-201, You have here an example of redundant sentence, because is similar to that in line 163. Another example is in line224 “Older patients are at higher risk of comorbidity and they often suffer from more than one medical condition” where authors express the same idea that in lines 186-187.“Moreover, the elderly tend to have other comorbidities, which increases their risk for hospitalization and readmission”.
- We have now rephrased most of the discussion to remove any redundant sentences as per the reviewer comment.
Line 205 “Moreover, patients with haematologic malignancies at high risk of hospitalization and for in-hospital death”. Is this afirmation for children too? Please specify what age group patients are you speaking about.
- We have now addressed this comment in page 15, lines 254-257.
Lines 212 “Colorectal cancer is frequently diagnosed at advanced stages when patients suffer from com213 plications”. I suggest to discuss here if preventive programs (occult blood in feces test screening/ preventive colonoscopy) in UK/Gales, are having some influence, supporting it with some reference.
- We have now addressed this comment in page 15, lines 268-279.
Line 221. “the decreasing incidence of bladder cáncer”. Could you give some factors to explain that decreasing.
- We have now addressed this comment in page 16, lines 296-299.
Line 230. Please define “SEER” (Surveillance, Epidemiology and End Results).
- We have now addressed this comment in page 16.
Line 238, Why talk about sex differences in cardiovascular diseases if cardiovascular is not the aim of your study? I see, more relevant, for example, the lung cancer evolution in females during the last years, which is approaching to the males rate (figure 3.a), long time after the incorporation of women to the smoking habit due to the lung cancer latency period. This is related to “gender” more than to “sex”.
- We discussed gender variation related to cardiovascular diseases as they share common risk factors with cancer including smoking and obesity.
Lines 257-259 I suggest some reference about to support this sentence. You have this one about eating habits
https://doi.org/10.1186/s12916-022-02256-w
- We have addressed the reviewer comment and added the recommended reference.
Strengths and Limitations
Line 275-276. “Other limitations include a lack of information on gender at the age-group level, rural/urban residence, and ethnicity for NRHA data”.
Authors mention in line 84 that “Available data include patient demographics…”. In this sense, I encourage you to explote some demographics data.
- Only age and gender were available and presented in our results.
Conclusion
Line 280-282. “Because cancer survivors are increasing and the population is aging in the UK, effective programs are needed to slow down or reverse the concerning trend in NRHA”. It seems to me an authors opinion more than a conclusion of your study.
- We have now rephrased the conclusion to address the reviewer comment.
References
Please, check and review the references, following an uniform style.
Some references don´t have the date when were accesed from internet.
- We will address all these issue with the production team. Thank you for your time and efforts in reviewing our manuscript which improved it significantly.